# Colorectal cancer-associated SNP rs17042479 is involved in the regulation of NAF1 promoter activity

Josephine B. Olsson[1,2], Marietta B. Gugerel[1], Stine B. Jessen[1,3], Jannie Jørgensen[1,2], Ismail Gögenur[3], Camilla Hansen[1], Lene T. Kirkeby[3], Jørgen Olsen[4], Ole B. V. Pedersen[2], Peter M. Vestlev[5], Katja Dahlgaard[1], Jesper T. Troelsen[1]*

**1** Department of Science and Environment, Roskilde University, Roskilde, Denmark, **2** Department of Clinical Immunology, Zealand University Hospital, Naestved, Denmark, **3** Center for Surgical Science, Enhanced Perioperative Oncology (EPeOnc) Consortium, Department of Surgery, Zealand University Hospital, Køge, Denmark, **4** Faculty of Health and Medical Sciences, Copenhagen University, Copenhagen, Denmark, **5** Department of Oncology, Roskilde Hospital, Roskilde, Denmark

* troelsen@ruc.dk

**Data Availability Statement:** All relevant data are within the paper and its Supporting Information files.

**Funding:** OBVP and JBO received funding from Region Sjællands Sundhedsvidenskabelige

## Abstract

A novel risk locus at 4q32.2, located between the *Nuclear Assembly Factor 1 (NAF1)* and *Follistatin Like 5 (FSTL5)* genes, was associated with increased risk of developing colorectal cancer (CRC), with SNP rs17042479 being the most associated. However, the link between CRC development and the risk locus at 4q32.2 is unknown. We investigated the promoter activity of *NAF1* and *FSTL5* and analyzed the risk locus at 4q32.2 as gene regulatory region. Our results showed that the activity of the *FSTL5* promoter was low compared to the *NAF1* promoter. Analyses of the *NAF1* promoter in conjunction with the region containing the risk locus at 4q32.2 showed that the region functions as gene regulatory region with repressor activity on *NAF1* promoter activity. The SNP rs17042479(G) increased the repressor effect of the region. CRC patients' biopsies were genotyped for SNP rs17042479(A/G), and *NAF1* expression profiles were examined. We found an association between SNP rs17042479 (G), cancer stage and tumor location. Additionally, patients with SNP rs17042479(G) showed lower *NAF1* expression in comparison to patients with SNP rs17042479(A) in tumor tissue and the *NAF1* expression in tumor tissue was lower compared to healthy tissue. The results in the study imply that reduced *NAF1* expression in the tumor contribute to a more aggressive phenotype. Furthermore, this study suggests that the SNP rs17042479(G) change the expression of *NAF1* and thereby increases the risk of developing CRC.

## Introduction

The third most diagnosed cancer worldwide is colorectal cancer (CRC). Additionally, CRC is the second most prevalent cause of cancer mortality [1]. A twin study estimated that 35% of CRC incidences are attributable to an inherited form of CRC [2]. Well-characterized inherited mutations, such as Lynch syndrome, familial adenomatous polyposis (FAP) and MUTYH-

Forskningsfond (RSSF) (R19A255B83). JBO
received funding from Region Syddanmarks og
Region Sjællands fælles forskningspulje (A170).
The funders had no role in study design, data
collection and analysis, decision to publish, or
preparation of the manuscript.

**Competing interests:** The authors have declared
that no competing interests exist.

associated polyposis (MAP) [3], are causing approximately 5% of all CRC cases [3–5]. The described well-characterized inherited mutations are all associated with high risk of CRC, with lifetime risk from 40% to 100% [3]. The remaining inherited CRC cases could be due to variations or polymorphisms more common, although less crucial, than the well-characterized mutations, but this is still incompletely understood [4,6]. Diagnosing patients earlier is desirable to obtain a better survival rate. Determining risk variations to identify individuals having a higher risk of developing CRC is therefore crucial [4]. A novel CRC risk locus at 4q32.2 was identified in 2014 from a genome-wide association study (GWAS). Six single nucleotide polymorphisms (SNP)s in the risk locus at 4q32.2 spanning a region of approximately 16.000 bp (chr4:163,325,384–163,341,245) were found to be associated with significantly higher risk of developing CRC. The six identified SNPs were: rs17042479 (A/G), rs79783178 (-/AT), rs35509282 (A/T), rs11736440 (A/G), rs9998942 (C/T) and rs57336275 (C/T). The risk locus at 4q32.2 is located between the two genes *Nuclear Assembly Factor 1 (NAF1)* and *Follistatin-Like 5 (FSTL5)* approximately 240kb upstream of the *FSTL5* gene and approximately 720kb downstream of the *NAF1* gene [6]. The coding region of *NAF1* gene is approximately 40.000 bp and include 8 exons. The coding region of *FSTL5* gene is approximately 725.000 bp and include 14 exons. NAF1 is involved in the assembly and accumulation of H/ACA ribonucleoproteins (RNPs) [7]. RNPs are enzymes, which contain a structurally conserved catalytic RNA molecule and highly divergent RNA-binding proteins [8]. H/ACA RNPs are one type of RNPs. To function, the H/ACA RNAs interact with a core set of proteins, which form the H/ACA RNP [9]. The H/ACA core proteins are GAR1, NHP2, NOP10 and NAP57 [7,10]. NAF1 is essential in the assembly of H/ACA RNPs, and the stable accumulation of H/ACA RNAs, together with three out of the four core proteins—NHP2, NOP10 and NAP57. H/ACA RNPs are necessary for telomere synthesis, modification of spliceosomal small nuclear RNAs, and ribosome biogenesis [10]. Wei *et al.* demonstrated that *NAF1* acted as an oncogene in glioma cells [11]. FSTL5 is a protein that forms complexes with activin. When FSTL5 is bound to activin, the actions of activin are neutralized. FSTL5 is involved in cell differentiation and embryogenesis [12]. Zhang *et al.* demonstrated that a downregulation of FSTL5 resulted in increased cell proliferation and survival in hepatocellular carcinoma [13].

The aim of this study was to investigate the association between the risk locus at 4q32.2 and the increased risk of developing CRC. We hypothesized that the risk locus at 4q32.2 affects the expression of *NAF1* or *FSTL5*, and that the changed NAF1 and/or FSTL5 expression influence the development of CRC. Through promoter reporter assays, the risk locus at 4q32.2 was analyzed as a putative gene regulatory region. Furthermore, we investigated a clinical dataset analyzing details about CRC patients' cancer characteristics and their *NAF1* expression in tumor and healthy tissue, as well as their genotype at the risk locus at 4q32.2, SNP rs17042479.

## Results

### Bioinformatic analysis of the risk locus at 4q32.2

The CRC risk locus at 4q32.2 identified by Schmit et al. [6] is located between the genes *NAF1* and *FSTL5* (Fig 1A). The association between the risk locus at 4q32.2 and the increased risk of developing CRC could be caused by the risk locus at 4q32.2 being located in a gene regulatory region. We examined previously published cap analysis of gene expression (CAGE) data (accession number: GSE95437) [14] to analyze whether the risk locus could be found in a gene regulatory region (Fig 1A). We found no transcriptional activity determined by CAGE around the *FSTL5* promoter, conversely, there was clearly transcriptional activity around the *NAF1* promoter. Furthermore, Fig 1A shows transcriptional activity in the risk locus at 4q32.2. This could indicate that the risk locus at 4q32.2 could be located in a gene regulatory region, as it has been shown that active gene regulatory regions produces unspecific transcription [15].

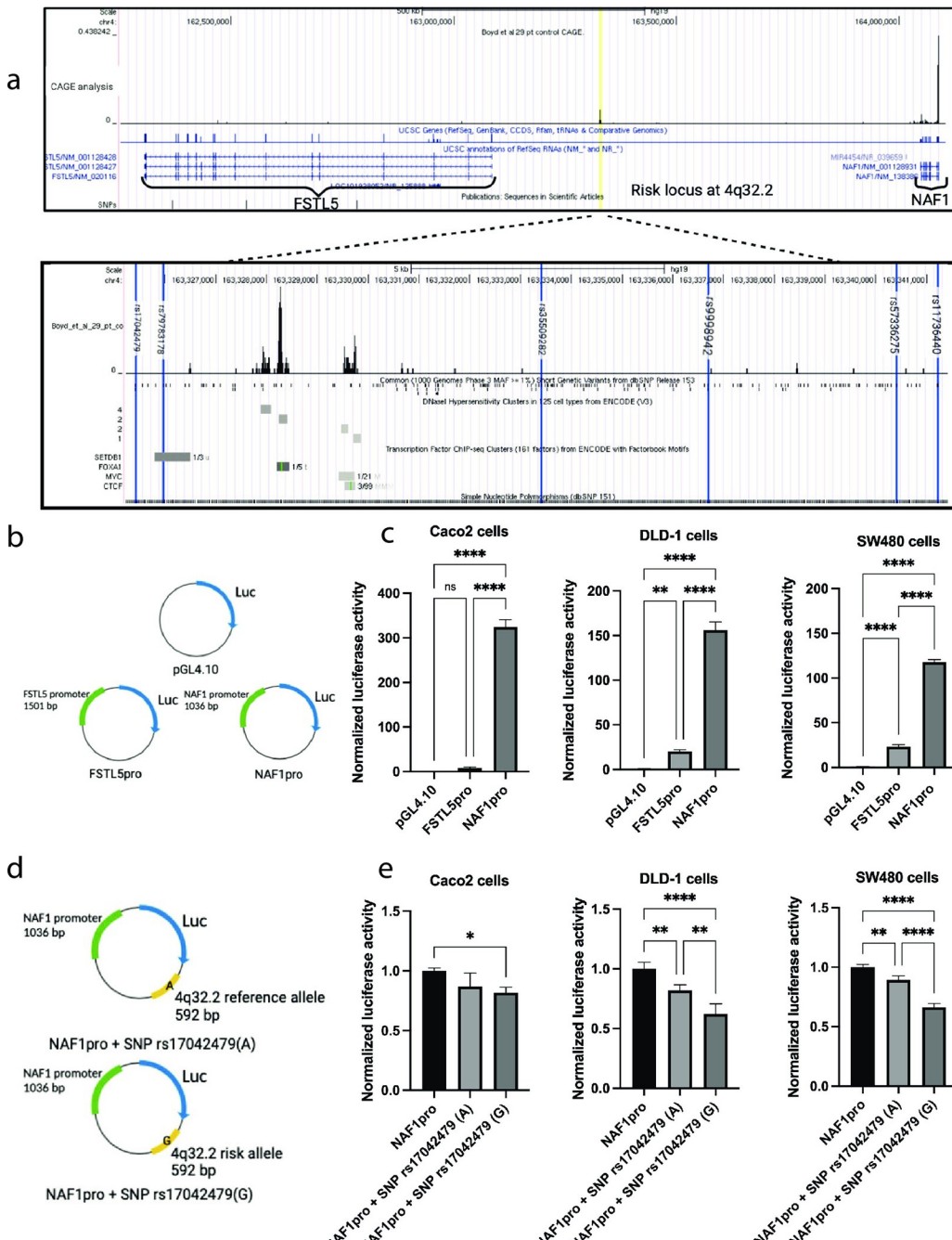

**Fig 1.** **(a)** Location of the risk locus at 4q32.2. CAGE analysis [14] of the region containing the two genes *NAF1* and *FSTL5* and the risk locus at 4q32.2, shown in yellow, are presented in the top [16]. The bottom presents a zoom in of the risk locus at 4q32.2, that shows the 6 SNPs found in Schmit *et al.*, highlighted in blue. From left: rs17042479 (A/G), rs79783178 (-/AT), rs35509282 (A/T), rs11736440 (A/G), rs9998942 (C/T) and rs57336275 (C/T) **(b)** Plasmid map of the construct used in the promoter reporter gene assay of promoter activity **(c)** Promoter reporter assay. Promoter activity of *FSTL5* (FSTL5pro) and *NAF1* (NAF1pro) in colon cancer cell lines. From left to right: Caco2 cells, DLD-1 cells and SW480 cells. Luciferase activities were corrected for transfection efficiency and normalized to the activity of the pGL4.10 without promoter (pGL4.10). N = 4. Statistical significance was determined using one-way ANOVA. **(d)** Plasmid map of the construct used in the promoter reporter gene assay of the gene regulatory region **(e)** Promoter reporter assay. The luciferase activities were corrected for transfection efficiency and normalized according to the expression of the *NAF1* promoter (NAF1pro). From left to right: Caco2 cells, DLD-1 cells and SW480 cells. NAF1pro + SNP rs17042479(A) is the construct with the *NAF1* promoter and the reference allele of SNP rs17042479. NAF1pro + SNP rs17042479(G) is the construct with the *NAF1* promoter and the risk allele of SNP rs17042479. N = 4. Statistical significance was determined using one-way ANOVA.

## Promoter activity of *NAF1* and *FSTL5* in colon cancer cells

We analyzed the promoter activity of *NAF1* and *FSTL5* in the three colon cancer cell lines Caco2, DLD-1 and SW480 in a promoter reporter gene assay (Fig 1C). The results of the promoter reporter assay demonstrated that the *FSTL5* promoter, FSTL5pro showed a slightly higher reporter gene activity in the two colon cancer cell lines DLD-1 and SW480 compared to background (the promoter-less pGL4.10 luciferase reporter plasmid). The *FSTL5* promoter, FSTL5pro, reporter gene activity was not significantly changed from background (the promoter-less pGL4.10 luciferase reporter plasmid) in Caco2 cells. Additionally, the *NAF1* promoter, NAF1pro, was highly active in all three colon cancer cell lines Caco2, DLD-1 and SW480. The promoter activity analysis revealed that the *FSTL5* promoter was not nearly as active in the three colon cancer cell lines as the *NAF1* promoter. As the CAGE data [14] (Fig 1A) also showed a very low transcriptional activity of the *FSTL5* promoter region *in vivo*, we conclude that any possible gene regulatory activity of the risk locus at 4q32.2 is most likely to take place on the *NAF1* promoter, and we therefore focused our further analyses on the *NAF1* promoter.

## The possible gene regulatory region in the risk locus at 4q32.2 has an impact on the *NAF1* promoter activity

A promoter reporter analysis was performed to determine if the risk locus at 4q32.2 functions as a gene regulatory region on the *NAF1* promoter activity, and whether the SNP rs17042479 (the SNP most significantly associated with CRC [6]) modifies such gene regulatory activity. Therefore, two versions of the risk locus were cloned into the *NAF1* promoter construct, NAF1pro. One containing the reference SNP rs17042479 (A) and one containing the risk SNP rs17042479(G). Analyzing the *NAF1* promoter activity in different constructs yielded similar results in the three colon cancer cell lines: Caco2, DLD-1, and SW480 (Fig 1C). The possible gene regulatory region with the reference allele SNP rs17042479 (A) significantly reduced the promoter activity of *NAF1* by 18% and 11% in the two colon cancer cell lines DLD-1 and SW480, respectively. In Caco2 cells, the possible gene regulatory region with the reference allele SNP rs17042479 (A) had no effect on *NAF1* promoter activity. The possible gene regulatory region with the risk allele SNP rs17042479(G) reduced the *NAF1* promoter activity by 18%, 38% and 34% in the three colon cancer cell lines Caco2, DLD-1 and SW480, respectively. In conclusion, it was demonstrated that the gene regulatory region with the risk SNP rs17042479 (G) has an increased repressor effect on the promoter activity of *NAF1* and is significantly different from the gene regulatory region with the reference allele SNP rs17042479 (A) in DLD-1 cells and SW480 cells.

The results from the bioinformatic analysis of the risk locus at 4q32.2 and the promoter reporter analysis indicated an association between the risk locus at 4q32.2 and the increased risk of developing CRC. This could be caused by a gene regulatory region containing SNP rs17042479, located in the risk locus at 4q32.2, which alters the expression of *NAF1*. To investigate the effect of SNP rs17042479 on cancer characteristics and the *NAF1* expression, along with the *NAF1* expression impact on cancer characteristics we analyzed a clinical dataset from patients diagnosed with CRC.

## Statistical analysis of the association between SNP rs17042479, *NAF1* expression and cancer characteristics

In order to investigate the possible role of the risk locus at 4q32.2 in CRC, we analyzed the relationship between the SNP rs17042479 (A/G), *NAF1* expression and cancer characteristics in

well-characterized biobank containing data and samples from 237 CRC patients [17]. The bio-bank holds information about patient age, gender, tumor differentiation grade, cancer stage, MMR status, tumor location, relapse, and death from other causes within a follow-up period of 5 years (1827 days). The *NAF1* and *Beta-2 Microglobulin (B2M)* expression data from CRC patients were determined from tumor and healthy intestinal tissue samples, and whether the patients were genotyped for the SNP rs17042479. 166 patients had the reference allele of SNP rs17042479(A), 44 patients had the risk allele of SNP rs17042479(G), and 27 of the patients were not genotyped. An overview of the cancer characteristics for the two genotype groups is presented in Table 1.

## The SNP rs17042479(G) and cancer characteristics

We analyzed whether the SNP rs17042479(G) is associated with cancer stage, tumor differenti-ation grade, tumor location and/or MMR-status. A statistically significant association was found between SNP rs17042479(G) and cancer stage (Fig 2A) and tumor location (Fig 2B). Patients with the risk SNP rs17042479(G) were significantly associated to be diagnosed at a later cancer stage compared to patients with the reference SNP rs17042479(A). SNP rs17042479(G) was also significantly associated with tumor location where patients harboring the risk allele of SNP rs17042479(G) more often had a right-sided colon cancer compared to patients with the reference allele of SNP rs17042479 (A).

No correlation was detected between the presence of SNP rs17042479(G) and differentia-tion grade and MMR-status.

**Table 1. Overview of cancer characteristics for patients homozygotic for the reference allele (Patients rs17042479 (A)) and for patients heterozygotic for the risk allele of SNP rs17042479 (Patients rs17042479(G)).** One patient was homozygotic for the risk allele.

| Characteristics | Patients rs17042479(A) (n = 166) | Patients rs17042479(G) (n = 44) |
|---|---|---|
| **Age, average, [years]** | | |
| | 68 [43–88] | 70 [43–90] |
| | | |
| **Gender** | Men: 100 (60) | Men: 24 (55) |
| | Women: 66 (40) | Women: 20 (45) |
| **Tumor location** | | |
| | Right: 83 (51) | Right: 19 (44) |
| | Left: 79 (49) | Left: 24 (56) |
| | | |
| **Tumor differentiation grade** | Poor: 39 (24) Moderate: 85 (51) Well: 42 (25) | Poor: 13 (30) Moderate: 23 (52) Well: 8 (18) |
| **Cancer stage** | I: 19 (11) II: 86 (52) III: 56 (34) IV: 5 (3) | I: 3 (7) II: 18 (41) III: 17 (38) IV: 6 (14) |
| **MMR status** | Deficient: 40 (24) Proficient: 125 (76) | Deficient: 8 (18) Proficient: 36 (82) |
| **Relapse** | Yes: 28 (17) No: 120 (72) | Yes: 5 (11) No: 35 (80) |
| **Death from other causes during follow-up** | 18 (11) | 4 (9) |
| **Average follow-up time, days [range]** | 1156 [5–1827] | 1091 [24–1827] |

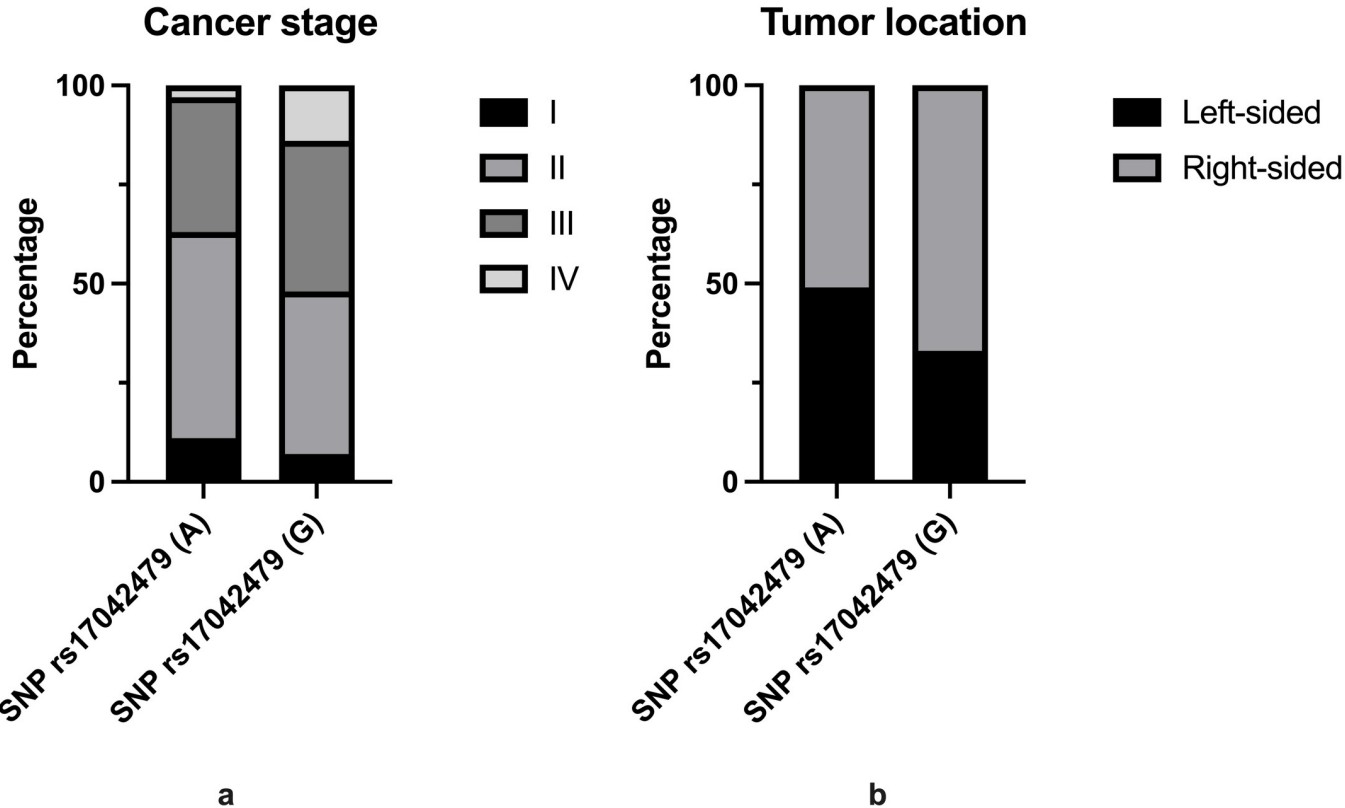

**Fig 2. (a)** Barplot of SNP rs17042479(G) distribution in cancer stages. The analysis was examined by Chi2-test and Fisher's exact test; p-value = 0.02. **(b)** Barplot of SNP rs17042479(G) distribution in tumor location. The analysis was examined by Chi2-tests and Fisher's exact test; p-value = 0.03.

## Association between SNP rs17042479(G) and *NAF1* expression in healthy and tumor tissue

We investigated if the *NAF1* expression differed between tumor and healthy tissue (Fig 3A). In addition, we analyzed if there was an association between the presence of risk SNP rs17042479 (G) and the *NAF1* expression in healthy and tumor tissue (Fig 3B and 3C).

In general, we found that *NAF1* expression was lower in tumor tissue compared to healthy tissue (Fig 3A). No correlation was found between the *NAF1* expression and the presence of the SNP rs17042479(G) in healthy tissue (Fig 3B). Conversely, a correlation was found between the *NAF1* expression and the presence of the SNP rs17042479(G) in tumor tissue (Fig 3C). Patients harboring the risk allele of SNP rs17042479(G) had a significantly lower relative *NAF1* expression in tumor tissue compared to patients with the reference allele of SNP rs17042479(A) (Fig 3C).

## Discussion

Early diagnosis of CRC is essential for the patients' chances of survival and recovery [4]. Identifying individuals with higher risk of developing CRC would be a way to improve both CRC screening and preventive therapies and thereby survival and recovery rates [18]. Determining variations in genetic risk loci are a way to identify individuals at a higher risk of developing CRC. Genome-wide association studies (GWAS) are used to identify new genetic risk factors for common diseases in a population by analyzing DNA sequence variations [19,20]. The goal of GWAS is to use involved genetic risk factor(s) to develop prediction models, and thereby

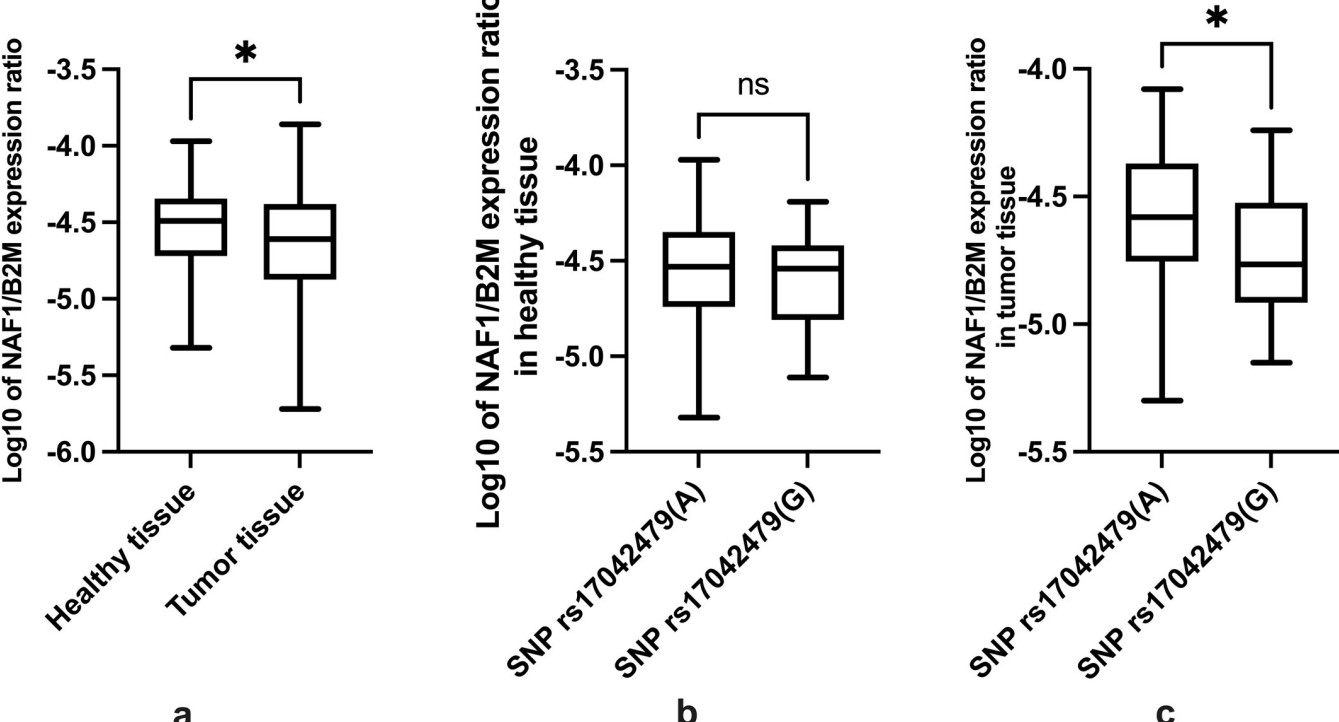

**Fig 3. (a)** Boxplot of *NAF1* expression in respectively healthy and tumor tissue. The patients with tumor content under 50% are not included in the analysis. n = 121. The log transformed data was examined by paired t-test; p-value = 0.012. **(b)** Boxplot of the *NAF1* expression in healthy tissue and the presence of SNP rs17042479(G). n = 43 for patients with the risk allele and n = 155 for patients with the reference allele. The analysis was examined by unpaired t-test; p-value = 0.28. **(c)** Boxplot of the *NAF1* expression in tumor tissue and the presence of SNP rs17042479. The patients with tumor content under 50% are not included in the analysis. n = 22 for patients with the alternative allele, and n = 99 for patients with the reference allele. The analysis was examined by unpaired t-test; p-value = 0.05.

identifying those that are at risk in the population. GWAS are widely used to associate genetic variants, such as SNPs, to a specific disease [18–20]. More than 70,000 variant-trait associations were found through GWAS in September 2018 [21]. When GWAS implicate genes, follow-up experiments are needed to discover the novel biological mechanisms that explains the link between the variant and the disease [18,22]. One of the primary goals of genetic research is to translate biological discoveries into medical advances. Even though it takes a lot of effort to translate biological discoveries into medical advances, it is a critical step, and there are more and more examples of GWAS findings with clinical applications [22]. Here, we address the underlying mechanisms why the risk locus at 4q32.2 identified by Schmit et al., [6] is associated with higher risk of developing CRC.

Two studies discovered a SNP 40kb downstream of *NAF1* that was linked to abnormal telomere length. NAF1 was found to be associated with longer telomere length by Walsh et al., whereas NAF1 was found to be associated with shorter telomere length by Stanley et al. [23,24]. These previous findings were not consistent, but it does suggest that NAF1 affects telomere length. CRC is associated with both shorter and longer telomere lengths, though shorter telomere lengths are more common [25].

### The *NAF1* promoter is highly active in colon cancer cell lines and in the colon epithelium

The *NAF1* promoter was highly active in the three colon cancer cell lines: SW480, DLD-1 and Caco2 (Fig 1C). The *FSTL5* promoter was also more active than the pGL4.10 luciferase

reporter plasmid in the two colon cancer cell lines DLD-1 and SW480, but not in Caco2 cells (Fig 1C). The *FSTL5* promoter was, however, not nearly as active as the *NAF1* promoter. Additionally, when analyzing the CAGE expression data from colon biopsies [14], there were no detectable transcriptional activity at the *FSTL5* promoter, whereas the *NAF1* promoter region was clearly active (Fig 1A). This led us to further investigate the impact of the risk locus at 4q32.2 on the *NAF1* promoter activity.

## The gene regulatory region in the risk locus at 4q32.2 showed repressor activity on the *NAF1* promoter

If the risk locus at 4q32.2 functions as a gene regulatory region, it has the potential to alter the expression of a gene that influences the development of CRC, which could explain the link between the risk locus at 4q32.2 and the increased risk of developing CRC. We analyzed whether the risk locus changes the activity of the *NAF1* promoter through a promoter reporter gene assay. The promoter reporter gene analysis showed that the region in the risk locus at 4q32.2 functions as a repressor on the *NAF1* promoter (Fig 1E). Signifying the risk locus at 4q32.2 potential as a gene regulatory region and providing an explanation for why the risk locus at 4q32.2 is linked to CRC. The risk locus was identified in a GWAS [6], where the SNP rs17042479 was identified as the most significant SNP associated to CRC. We therefore investigated if the SNP rs17042479(G) changes the activity of the gene regulatory region and found that it significantly increased the repressor effect of the region in the risk locus at 4q32.2. This indicated that the link between the 4q32.2 risk locus and CRC could be caused by gene regulatory activity altering the activity of the *NAF1* promoter. Although promoter reporter gene assays have made significant contributions to the analysis of eukaryotic gene expression and regulation [26], the promoter reporter assay has the limitation that the DNA in the plasmid is not organized in a chromatin structure like the genomic DNA in the cell. As a result, the promoter reporter assay analyzes promoter activity without taking into account the influence of epigenetic mechanisms. Epigenetics is important in the regulation of gene expression, often in a close interplay with transcriptional regulators [27]. Patients with the risk allele of SNP rs17042479 included in this study is heterozygous for the SNP rs17042479, it is possible that the effect would be more evident if the patients included were homozygous for the risk allele of SNP rs17042479.

## The SNP rs17042479(G) is associated with cancer stage and tumor location

We found that the risk SNP rs17042479(G) was associated with diagnosis at a later cancer stage (Fig 2A). This could imply that the SNP rs17042479(G) is associated with a more aggressive CRC development, resulting in patients with the risk SNP rs17042479(G) being diagnosed at a later cancer stage. It could also be explained by the finding that patients with the risk SNP rs17042479(G) were more likely to develop right-sided colon cancer than left-sided colon cancer, compared to the patients with the reference allele of SNP rs17042479(A) (Fig 2B). Several studies have investigated the prognostic impact of tumor location [28–30]. There are a number of differences between the two tumor locations; epidemiology, clinical presentation, pathology and genetic mutations [29]. Patients with right-sided CRC are associated with worse prognosis compared to patients with left-sided CRC, and are also associated with more advanced tumors, which are less differentiated [28–30].

## Lower tumor *NAF1* expression in patients with SNP rs17042479(G)

The association between SNP rs17042479(G) and the increased risk of developing CRC could be explained by a changed expression of *NAF1* mediated through the gene regulatory effect of

SNP rs17042479(G) affecting cancer characteristics. The patients with the risk SNP rs17042479(G) were associated with lower expression of *NAF1* compared to patients with the reference SNP rs17042479(A) in tumor tissue (Fig 3C). The *NAF1* expression in healthy tissue was not altered between the two genotype groups. The observation that the risk allele of SNP rs17042479 (G) seems to have different impact in healthy tissue compared to malignant tissue could indicate cancer-related transcriptions factors binds better to the risk allele compared to the reference allele. An example of a transcription factor with increased activity in colon cancer cells is TCF7L2 (TCF4), the main transcription factor activated by Wnt signaling. The Wnt signaling pathway is a key pathway in colorectal cancer pathogenesis and is constitutively active in APC mutated colon cancer cells [31]. A TCF7L2 binding site has previously been described in an enhancer of the MYC gene [32]. A colon cancer-associated single nucleotide variant in this binding site increases the binding of TCF7L2 and the expression of MYC. As most studies of gene regulatory activity is done in cell lines; it could be an overlooked phenomenon that nucleotide variants can have different effects on gene regulatory elements in normal and diseased cells. The *NAF1* expression in tumor tissue was lower than the *NAF1* expression in healthy tissue (Fig 3A). This suggests that low *NAF1* expression in tumor tissue is associated with a poor prognosis for CRC patients. It would have been interesting to investigate a survival analysis of the two genotype groups and high and low *NAF1* expression, but due to the size of the dataset this was not possible. Human Protein Atlas (https://www.proteinatlas.org) has analyzed if the survival rate of CRC was affected by the NAF1 expression in a Kaplan Meier plot. The 597 patients included in the Kaplan Meier analysis was divided into the two groups high and low NAF1 expression. The survival analysis showed a difference in survival rate between the two groups high NAF1 expression (5-year survival: 73%) and low NAF1 expression (5-year survival: 57%) [33].

A limitation in this study is the relatively small size of the patient group analyzed. The group of patients with the risk allele of SNP rs17042479 is rather small (n = 44). Furthermore, it would have been optimal to have a group of patients that is heterozygous for the risk allele of SNP rs17042479 and a group that is homozygous for the risk allele of SNP rs17042479. Furthermore, the promoter reporter assays do not fully mimic the chromosomal regulation of a gene as the transfected plasmid DNA is not organized as chromatin, probably excluding epigenetic mechanisms in the analysis.

Wei *et al.*, demonstrated in 2019 an association between NAF1 and gliomas. Conversely, from the results in this study, Wei *et al.* found an association between high expression of NAF1 and poor patient survival. The results from Wei *et al.*, revealed that *NAF1* functions as an oncogene in glioma cells, by promoting cell growth. In this study, the results demonstrated that *NAF1* functions as a tumorsuppressorgene in CRC. Gliomas and CRC are two very different cancer types, and it is therefore possible that NAF1 functions differently in the two cancer types.

Our study implicates that *NAF1* could act as a tumorsuppressorgene in CRC. Lower *NAF1* expression in the tumors drives the cancer towards a more aggressive phenotype because the risk SNP rs17042479(G) is associated with metastasizing tumors (stage 4) as well as right-side location of the tumor. We suggest that the SNP rs17042479(G) increases the risk of developing CRC by altering the promoter activity of *NAF1*.

## Materials and methods

### Cell culture

DLD-1, Caco-2 and SW480 cells were used in this study. DLD-1 cells were grown in McCoy's M5 medium with L-Glutamine (biowest or Lonza), 10% Fetal Bovine Serum (FBS) (HyClone)

and 1% Penicillin/Streptomycin (PS) (Lonza). Caco-2 and SW480 cells were grown in Dulbecco's modified essential medium (DMEM) with L-Glutamine (Lonza), 10% FBS (HyClone) and 1% PS (Lonza). All cell lines were regularly sub-cultivated and incubated at 37˚C, 5% CO2.

## Constructs to promoter reporter assay

The *NAF1* promoter (chr4: 164,087,998–164,089,033 in GRCh37/hg19) was cloned into the restriction site of HinDIII and the *FSTL5* promoter (chr4:163,084,882–163,086,382 in GRCh37/hg19) was cloned into the restriction sites of XhoI and BglII in the pGL4.10 [luc2] vector. The possible gene regulatory region (chr4: 163,325,126–163,325,717 in GRCh37/hg19) was cloned into the restriction sites of BamHI and SalI in the pGL4.10 [luc2] vector. The construct with *NAF1* promoter and gene regulatory region with the reference allele of SNP rs17042479 were made by site directed mutagenesis [34]. The remaining constructs were made from purchased human genomic DNA (Roche Diagnostics GmbH) and the In-Fusion cloning strategy [35]. The In-Fusion cloning was performed following the manufacturer's protocol [34,35]. All promoter reporter constructs were confirmed by sequencing.

## Promoter reporter assay

Cells were seeded in 24-wells plates at a concentration of 4·104 cells per well. Four replicas were made of each of the constructs with a total DNA concentration of 1.2 μg containing 0.2 μg construct, 0.1 μg CMV LacZ and 0.9 μg pSK+. 100 μl 2μM Polyethylenimine (PEI) (Alfa Aesar) diluted in 150 mM NaCl were added to each of the 100 μl DNA mixtures. The PEI/DNA solutions were incubated for 1 hour at room temperature. 49 μl of each mixture was added in small drops on each of the 4 replicas. The plates were spun at 1200 rpm for 5 minutes and incubated for 4 hours at 37˚C. After 4 hours the medium was removed and new medium was added, and the plates were afterwards incubated at 37˚C. The luciferase and β-galactosidase were measured two days after transfection. Before the measurements the cells were washed with 200 μl Phosphate buffered saline (PBS) (Sigma Aldrich) and then lysed with 130 μl lysis buffer mixed with Dithiothreitol (DTT) (Sigma Aldrich) at a concentration of 0.5 mM. 10 μl of each sample was transferred to a white 96-well plate [36]. The Dual-Light system was used containing both Buffer A (Applied Biosystems), Buffer B (Applied Biosystems) with 1:1000 Galacton-Plus (Applied Biosystems), and Accelerator II (Applied Biosystems). The assay was run using the GloMax Luminometer. 25 μl Buffer A, 100 μl Buffer B and 100 μl Accelerator II were used for each 10 μl sample. The GloMax luminometer was programmed to add Buffer A and B simultaneously and after 2 seconds measure the luciferase activity for 5 seconds. The β-galactosidase activity was measured 45 minutes after the first luciferase measurement. 2 seconds after addition of Accelerator II the β-galactosidase activity was measured for 5 seconds. The promoter reporter gene analysis was repeated three times with the same pattern; however, only one representative repetition is shown in the results section.

## Colon cancer patient analyses

The patient material in the clinical data was compiled from the cancer Biobank at the University of Copenhagen [37]. The clinical data were collected between September 2006 and May 2012 from Roskilde University Hospital [17]. The patients' NAF1 expression were determined by qPCR (primers: Forward: CACCACCAGAGGCCTTAGAT, reverse: CCATGGCAAGATC-GAGGGTA) and the SybrGreen kit (Roche lifescience) was used. The *NAF1* were normalized to expression of *B2M*. *B2M* has previously been identified as an good pPCR reference gene studying human colon carcinomas score [38]. The majority of the patients included in the clinical

dataset were also genotyped for SNP rs17042479 using the SimpleProbe genotyping method, and a predesigned LightSNiP assay (TIBMOLBIO).

## Data source

The study was approved by the Danish Committee on Health Research and Ethics, Region Zealand (protocol no: SJ-373) and the Danish Regional Data Protection Agency (File no: REG-072-2018). Written informed consent has been obtained from the patient(s) to publish this paper.

## Statistical analysis

The data was analyzed and statistical significance was determined using Graphpad Prism 9.1.1. Categorical data was examined with Chi2-tests and Fisher's exact test. Numerical data was examined by t-test. The statistical test is determined significant if $P \leq 0.05$, furthermore we use following symbols for different significance levels: $^*P<0.05$, $^{**}P<0.01$, $^{***}P<0.001$ and $^{****}P<0.0001$.

## Supporting information

**S1 Dataset.**
(XLSX)

## Acknowledgments

We would like to thank Jesper Olsen for providing us the clinical dataset used in the statistical analysis. Jesper Olsen extracted RNA and synthesized cDNA [17]. Furthermore, we would like to thank Marianne Lauridsen and Lotte Laustsen for assisting in the laboratory work.

## Author Contributions

**Conceptualization:** Josephine B. Olsson, Marietta B. Gugerel, Stine B. Jessen, Katja Dahlgaard, Jesper T. Troelsen.

**Data curation:** Josephine B. Olsson, Marietta B. Gugerel, Jannie Jørgensen, Camilla Hansen, Jørgen Olsen, Peter M. Vestlev, Katja Dahlgaard, Jesper T. Troelsen.

**Formal analysis:** Josephine B. Olsson, Marietta B. Gugerel, Camilla Hansen, Jørgen Olsen, Peter M. Vestlev, Katja Dahlgaard, Jesper T. Troelsen.

**Funding acquisition:** Josephine B. Olsson, Ole B. V. Pedersen, Katja Dahlgaard, Jesper T. Troelsen.

**Investigation:** Josephine B. Olsson, Marietta B. Gugerel, Lene T. Kirkeby, Jørgen Olsen, Katja Dahlgaard, Jesper T. Troelsen.

**Methodology:** Josephine B. Olsson, Marietta B. Gugerel, Stine B. Jessen, Jannie Jørgensen, Ismail Gögenur, Lene T. Kirkeby, Ole B. V. Pedersen, Peter M. Vestlev, Katja Dahlgaard, Jesper T. Troelsen.

**Project administration:** Katja Dahlgaard, Jesper T. Troelsen.

**Resources:** Josephine B. Olsson, Lene T. Kirkeby, Ole B. V. Pedersen, Katja Dahlgaard, Jesper T. Troelsen.

**Software:** Josephine B. Olsson, Marietta B. Gugerel.

**Supervision:** Ismail Gögenur, Ole B. V. Pedersen, Katja Dahlgaard, Jesper T. Troelsen.

**Validation:** Josephine B. Olsson, Marietta B. Gugerel, Katja Dahlgaard, Jesper T. Troelsen.

**Visualization:** Josephine B. Olsson, Jesper T. Troelsen.

**Writing – original draft:** Josephine B. Olsson.

**Writing – review & editing:** Josephine B. Olsson, Marietta B. Gugerel, Stine B. Jessen, Jannie Jørgensen, Ismail Gögenur, Jørgen Olsen, Ole B. V. Pedersen, Katja Dahlgaard, Jesper T. Troelsen.

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
