## [Decision Letter · Decision Letter 0]

13 Apr 2022

PONE-D-22-08015Colorectal cancer-associated SNP rs17042479 is involved in the regulation of NAF1 promoter activityPLOS ONE

Dear Dr. Thorvald Troelsen,

Thank you for submitting your manuscript to PLOS ONE. After careful consideration, we feel that it has merit but does not fully meet PLOS ONE’s publication criteria as it currently stands. Therefore, we invite you to submit a revised version of the manuscript that addresses the points raised during the review process.

We look forward to receiving your revised manuscript.

Kind regards,

Alvaro Galli

Academic Editor

PLOS ONE

Journal Requirements:

Reviewers' comments:

Reviewer's Responses to Questions

**Comments to the Author**

1. Is the manuscript technically sound, and do the data support the conclusions?

Reviewer #1: Partly

Reviewer #2: Partly

2. Has the statistical analysis been performed appropriately and rigorously? 

Reviewer #1: No

Reviewer #2: Yes

3. Have the authors made all data underlying the findings in their manuscript fully available?

Reviewer #1: Yes

Reviewer #2: Yes

4. Is the manuscript presented in an intelligible fashion and written in standard English?

Reviewer #1: Yes

Reviewer #2: Yes

5. Review Comments to the Author

Reviewer #1: Dear Respected authors,

Please find my comments about your manuscript below for your perusal and revision of the paper:

In abstract the conclusion lines should be clear and future driven. No such take home message is carried in the current abstract.

It would be better if the author can include the prediction analysis of the said new SIX SNPS as to how they will affect the functioning of the two genes Nuclear Assembly Factor 1 (NAF1) and Follistatin-Like 5 (FSTL5).

Introduction section should add more information on the gene, their SNPs and the cross talk.

Please provide either graphical representation of the SIX SNPS or tabulated form, showing the location of each SNP in the gene and what substitutions do occur in each. The current graphical representation (Figure 1a) is not good enough as well.

The analysis of the cases based on grouping is also heterogeneous, which is important as it biases the results and interpretations implicating implicates that NAF1 as a tumor suppressor gene in CRC. Author need to make a clear explanation of this.

Author should exhaustively write the statistical analysis and test done. What were the descriptive and inferential statistics that were performed, need to be mention clearly?

It is not clear as to why Authors have kept four different values of P value as significant. The norm is always P<=0.05 is regarded as significant.

Authors need to enlist all limitations of the study that might bias the interpretations at the end of the discussion.

Reviewer #2: Thank you for the chance you gave me to read this interesting study entitled “Colorectal cancer-associated SNP rs17042479 is involved in the regulation of NAF1 promoter activity” by Olsson et al. In this original research study, the authors investigated the promoter activity of NAF1 and FSTL5, the risk locus at 4q32.2 as gene regulatory region as well as the significance of SNP rs17042479 in colorectal cancer patients. This is a very interesting topic and the study is well-written. However, I think that this study in the current form doesn’t satisfy the appropriate criteria for publication in your journal since there are some major points which need to be treated before publication.

Some major issues are:

1. A two-phase design would be more desirable and possibly, more robust results would have been drawn.

2. The authors should explain why B2M gene was chosen as reference gene. If they had used other housekeeping gene, maybe, presented results would have been more conclusive.

3. An important issue is also the small size of the cohort which may limit study significance.

4. Based on the presented results, SNP rs17042479 (G) seems to have different impact in healthy compared to malignant tissues. This is a weird finding. How this observation could be explained?

5. Why the authors used B2M as a reference gene? Are there supportive data on CRC?

6.Presented results regarding the associations of NAF1 expression with stage/differentiation seem to be arbitrary or borderline, especially if we take into consideration the number of patients included in stage IV and the extensive overlapping in boxplots (Fig 4).

7. Line 266-267: Please, this statement “Patients with the risk allele of SNP rs17042479 are heterozygous for the SNP, it is possible that the effect is more evident in patients that are homozygous for the risk allele of SNP rs17042479” should be clarified.

Minor issue:

Abbreviations should be expanded throughout the manuscript.

6. PLOS authors have the option to publish the peer review history of their article (what does this mean?). If published, this will include your full peer review and any attached files.

Reviewer #1: **Yes: **Syed Sameer Aga

Reviewer #2: No

---

## [Author Response · Author response to Decision Letter 0]

26 May 2022

We have addressed the editorial comments 1-4 and we have added our responses to the points and comments raised by the reviewers in the document response to reviewers.

---

## [Decision Letter · Decision Letter 1]

20 Jun 2022

PONE-D-22-08015R1Colorectal cancer-associated SNP rs17042479 is involved in the regulation of NAF1 promoter activityPLOS ONE

Dear Dr. Thorvald Troelsen,

Thank you for submitting your manuscript to PLOS ONE. After careful consideration, we feel that it has merit but does not fully meet PLOS ONE’s publication criteria as it currently stands. Therefore, we invite you to submit a revised version of the manuscript that addresses the points raised during the review process.

We look forward to receiving your revised manuscript.

Kind regards,

Alvaro Galli

Academic Editor

PLOS ONE

Journal Requirements:

Reviewers' comments:

Reviewer's Responses to Questions

**Comments to the Author**

1. If the authors have adequately addressed your comments raised in a previous round of review and you feel that this manuscript is now acceptable for publication, you may indicate that here to bypass the “Comments to the Author” section, enter your conflict of interest statement in the “Confidential to Editor” section, and submit your "Accept" recommendation.

Reviewer #2: All comments have been addressed

2. Is the manuscript technically sound, and do the data support the conclusions?

Reviewer #2: Yes

3. Has the statistical analysis been performed appropriately and rigorously? 

Reviewer #2: Yes

4. Have the authors made all data underlying the findings in their manuscript fully available?

Reviewer #2: Yes

5. Is the manuscript presented in an intelligible fashion and written in standard English?

Reviewer #2: Yes

6. Review Comments to the Author

Reviewer #2: The authors have adressed the majority of my comments with the exception of the comment regarding the different impact of SNP rs17042479 (G) in healthy compared to malignant tissues.

Although the authors have added a sentence in the discussion section, providing a possible explanation, however, they don't provide relevant refs which could support this explanation. This point needs to be treated further.

7. PLOS authors have the option to publish the peer review history of their article (what does this mean?). If published, this will include your full peer review and any attached files.

Reviewer #2: No

---

## [Author Response · Author response to Decision Letter 1]

31 Jul 2022

Dear editor and reviewer. 

Thank you very much for the extra comments. We have checked the reference list to ensure that it is complete and correct. 

Comments from reviewer 2:

The authors have adressed the majority of my comments with the exception of the comment regarding the different impact of SNP rs17042479 (G) in healthy compared to malignant tissues.

Although the authors have added a sentence in the discussion section, providing a possible explanation, however, they don't provide relevant refs which could support this explanation. This point needs to be treated further.

We have elaborated more on this finding in the discussion and added references.

---

## [Decision Letter · Decision Letter 2]

22 Aug 2022

Colorectal cancer-associated SNP rs17042479 is involved in the regulation of NAF1 promoter activity

PONE-D-22-08015R2

Dear Dr.Thorvald Troelsen,

We’re pleased to inform you that your manuscript has been judged scientifically suitable for publication and will be formally accepted for publication once it meets all outstanding technical requirements.

Kind regards,

Alvaro Galli

Academic Editor

PLOS ONE

Additional Editor Comments (optional):

Reviewers' comments:

Reviewer's Responses to Questions

**Comments to the Author**

1. If the authors have adequately addressed your comments raised in a previous round of review and you feel that this manuscript is now acceptable for publication, you may indicate that here to bypass the “Comments to the Author” section, enter your conflict of interest statement in the “Confidential to Editor” section, and submit your "Accept" recommendation.

Reviewer #2: All comments have been addressed

2. Is the manuscript technically sound, and do the data support the conclusions?

Reviewer #2: Yes

3. Has the statistical analysis been performed appropriately and rigorously? 

Reviewer #2: Yes

4. Have the authors made all data underlying the findings in their manuscript fully available?

Reviewer #2: Yes

5. Is the manuscript presented in an intelligible fashion and written in standard English?

Reviewer #2: Yes

6. Review Comments to the Author

Reviewer #2: Thank you for the chance you gave me to read this interesting study entitled “Colorectal cancer-associated SNP rs17042479 is involved in the regulation of NAF1 promoter activity” by Olsson et al.This is a very interesting topic and the study is well-written. All comments have been adressed. I think that this study in the current form satisfies the appropriate criteria for publication

7. PLOS authors have the option to publish the peer review history of their article (what does this mean?). If published, this will include your full peer review and any attached files.

Reviewer #2: No

---

## [Editor Report · Acceptance letter]

26 Aug 2022

PONE-D-22-08015R2 

Colorectal cancer-associated SNP rs17042479 is involved in the regulation of NAF1 promoter activity 

Dear Dr. Troelsen:

I'm pleased to inform you that your manuscript has been deemed suitable for publication in PLOS ONE. Congratulations! Your manuscript is now with our production department. 

Kind regards, 

on behalf of

Dr. Alvaro Galli 

Academic Editor

PLOS ONE